# A mechanistic and data-driven reconstruction of the time-varying reproduction number: Application to the COVID-19 epidemic

Bernard Cazelles[1,2,3]*, Clara Champagne[4,5], Benjamin Nguyen-Van-Yen[3,6], Catherine Comiskey[7], Elisabeta Vergu[2◉], Benjamin Roche[8◉]

**1** Sorbonne Université, UMMISCO, Paris, France, **2** INRAE, Université Paris-Saclay, MaIAGE, Jouy-en-Josas, France, **3** Eco-Evolution Mathématique, IBENS, UMR 8197, CNRS, Ecole Normale Supérieure, Paris, France, **4** Swiss Tropical and Public Health Institute, Basel, Switzerland, **5** Universty of Basel, Basel, Switzerland, **6** Institut Pasteur, Unité de Génétique Fonctionnelle des Maladies Infectieuses, Paris, France, **7** School of Nursing and Midwifery, Trinity College Dublin, The University of Dublin, Dublin, Ireland, **8** MIVEGEC, IRD, CNRS and Université de Montpellier, Montpellier, France

◉ These authors contributed equally to this work.
* cazelles@bologie.ens.fr

**Data Availability Statement:** The code and data used are available at: 10.5281/zenodo.5032469.

## Abstract

The effective reproduction number $R_{eff}$ is a critical epidemiological parameter that characterizes the transmissibility of a pathogen. However, this parameter is difficult to estimate in the presence of silent transmission and/or significant temporal variation in case reporting. This variation can occur due to the lack of timely or appropriate testing, public health interventions and/or changes in human behavior during an epidemic. This is exactly the situation we are confronted with during this COVID-19 pandemic. In this work, we propose to estimate $R_{eff}$ for the SARS-CoV-2 (the etiological agent of the COVID-19), based on a model of its propagation considering a time-varying transmission rate. This rate is modeled by a Brownian diffusion process embedded in a stochastic model. The model is then fitted by Bayesian inference (particle Markov Chain Monte Carlo method) using multiple well-documented hospital datasets from several regions in France and in Ireland. This mechanistic modeling framework enables us to reconstruct the temporal evolution of the transmission rate of the COVID-19 based only on the available data. Except for the specific model structure, it is non-specifically assumed that the transmission rate follows a basic stochastic process constrained by the observations. This approach allows us to follow both the course of the COVID-19 epidemic and the temporal evolution of its $R_{eff}(t)$. Besides, it allows to assess and to interpret the evolution of transmission with respect to the mitigation strategies implemented to control the epidemic waves in France and in Ireland. We can thus estimate a reduction of more than 80% for the first wave in all the studied regions but a smaller reduction for the second wave when the epidemic was less active, around 45% in France but just 20% in Ireland. For the third wave in Ireland the reduction was again significant (>70%).

**Funding:** BC and BR are partially supported by a grant ANR Flash Covid-19 from the "Agence Nationale de la Recherche" (DigEpi). The funder has no role in study design, data collection and analysis, decision to publish, or preparation of the manuscript.

**Competing interests:** The authors have declared that no competing interests exist.

## Author summary

In the early stages of any new epidemic, one of the first steps to design a control strategy is to estimate pathogen transmissibility in order to provide information on its potential to spread in the population. Among the different epidemiological indicators that characterize the transmissibility of a pathogen, the effective reproduction number $R_{eff}$ is commonly used for measuring time-varying transmissibility. It measures how many additional people can be infected by an infected individual during the course of an epidemic. However, $R_{eff}$ is difficult to estimate in the presence of silent transmission and/or significant temporal variation in case reporting. This is exactly the situation we are confronted with during this COVID-19 pandemic. The statistical methods classically used for the estimation of $R_{eff}$ have some shortcomings in the rigorous consideration of the transmission characteristics of SARS-CoV-2. We propose here to use an original approach based on a stochastic model whose parameters vary in time and are inferred in a Bayesian framework from reliable hospital data. This enables us to reconstruct both the COVID-19 epidemic and its $R_{eff}$. The $R_{eff}$ time evolution allows us to get information regarding the potential effects of mitigation measures taken during and between epidemics waves. This approach, based on a stochastic model that realistically describes the hospital multiple datasets and which overcomes many of the biases associated with $R_{eff}$ estimates, appears to have some advantage over previously developed methods.

## Introduction

In the last months of 2019, clustered pneumonia cases were described in China [1]. The etiological agent of this new disease, a betacoronavirus, was identified in January and named SARS-CoV-2. This new coronavirus disease (COVID-19) spread rapidly worldwide, causing millions of cases, killing hundreds of thousands of people and causing socio-economic damage. Until vaccination campaigns are widely implemented, the expansion of COVID-19 with the occurrence of new and more transmissible variants continues to threaten overwhelming the healthcare systems of many countries, despite a wide range of public health strategies using different non-pharmaceutical interventions (NPI).

In the early stages of each new epidemic, one of the first steps to design a control strategy is to estimate pathogen transmissibility in order to provide information on its potential to spread in the population. This is crucial to understand the likely trajectory of the new epidemic and the level of intervention that is needed to control it. Among the various indicators that quantify this transmissibility, the most commonly used is the reproduction number, which measures how many new infected individuals on average can be generated by one infected individual. In the initial phase of the epidemic, when the entire population is susceptible, this quantity is referred to as $R_0$, the basic reproduction number, and is defined as the average number of secondary cases caused by one infected individual in an entirely susceptible population [2,3]. As the epidemic develops and the number of infected individuals increases (and the number of susceptible individuals decreases), the effective reproduction number $R_{eff}$, characterizing the transmission potential according to the immunological state of the host population, is used instead. It can be estimated as a function of time, the instantaneous effective reproduction number $R_{eff}(t)$ quantifying the number of secondary infections caused by an infected person at a specific time-point of an epidemic. The epidemic is able to spread when $R_{eff}(t)>1$ and is under control when $R_{eff}(t)<1$. $R_{eff}(t)$ can be used to monitor changes in transmission in near real time and, as a consequence, for instance, of control or mitigation measures.

Classically, $R_{eff}(t)$ is estimated by the ratio of the number of new infections generated at time $t$ to the total infectiousness of individuals in the infected state at time $t$. This latter quantity is defined based on the generation time distribution (the time between infection and transmission) or on the serial interval (the time between the onset of symptoms of a primary case and the onset of symptoms of secondary cases) [4,5].

For the COVID-19 epidemic, considering the data needed for the estimation of the reproductive numbers, caution is urged when interpreting the values obtained and the short-term fluctuations in these estimates due to both data quality, which must be taken into account [6–9]. In a recent study by O'Driscoll et al [6], it was concluded, comparing different methods, that there are many important biases in the $R_{eff}(t)$ estimates and that this can easily lead to erroneous conclusions about changes in transmissibility during an epidemic. These biases are mainly due to the uncertainty in incidence data that can arise due to both the transmission characteristics of this virus (asymptomatic and pre-symptomatic transmission) and the quality and preparedness of the public health system. For COVID-19, it has been shown that the number of observed confirmed cases significantly underestimates the actual number of infections [10,11]. For instance, during the initial rapid growth phase of the COVID-19 epidemic, the number of confirmed case underestimated the actual number of infections by 50 to 100 times [10]. In France, it has been estimated that the detection rate increased from 7% in mid-May to 40% by the end of June, compared to well below 5% at the beginning of the epidemic [12]. In addition, these biases can be amplified by the combination of the high proportion of asymptomatic cases [13] with low health-seeking behaviors. Although the estimation of the reproductive number is robust to underreporting [14], this is only true if the reporting rate (among other characteristics) is constant over the observation period. This was not verified for the COVID-19 epidemic, mainly due to fluctuations in the capacity of testing and the near real-time availability of information. The uncertainty is not exclusively related to the incidence data, there is also a large variability in the value proposed in the literature for generation time or serial interval distributions and average values [15,16].

Other studies have highlighted these issues. Gostic et al [7] quantified the effects of data characteristics on $R_{eff}(t)$ estimates: reporting process; imperfect observation of cases; missing observations of recent infections; estimation of the generation interval. Moreover, Pitzer et al [8] showed that biases in $R_{eff}(t)$ are amplified when reporting delays have fluctuated due to the availability and changing practices of testing. These two studies concluded that changes in diagnostic testing and reporting processes must be monitored and taken into account when interpreting estimates of the reproductive number of COVID-19. Nevertheless these changes are extremely difficult to quantify. In these contexts, as testing capacity and reporting delays evolve, the use of hospital admission and mortality data may be preferable for inferring reproduction numbers [9,17]. However, the delays in the time from infection to hospitalization and/or death are also uncertain and, overall, it is difficult to incorporate the uncertainty related to all of these delays [9,17,18].

In addition to the classical methods, a complementary approach consists in inferring changes in transmission using mechanistic mathematical models, and computing $R_{eff}$ based on its proportionality with the transmission rates [19–24]. The time-variation of $R_{eff}$ is computed indirectly by simply fitting the model to different time periods (before or after the lockdown in the simplest cases) or by using exponential decay models [21], but see Lemaitre et al [20].

To overcome all these numerous weaknesses in the data available for estimating $R_{eff}$, we propose using a framework that has been already implemented to tackle non-stationarity in epidemiology [25–27]. It uses diffusion models driven by Brownian motion to model time-varying key epidemiological parameters embedded in a stochastic state-space framework coupled with Bayesian inference methods. The advantages of this approach compared to the

existing ones consist in (i) the description of the mechanisms underlying pathogen transmission and hence its particularities (asymptomatic phase), (ii) the joint use of multiple datasets (incidence and hospital data), (iii) the explicit taking into account of the uncertainty associated with the data used and especially (iv) the monitoring of the temporal evolution of some of the model parameters without inclusion of external variables. Overall, the main advantage of this approach is that it is data-driven. Indeed, in this framework, except for the mechanistic assumptions underlying the model, the estimation of the time-varying parameters, based on the available epidemiological observations [27], is done only under the non-specific assumption that they follow a basic stochastic process.

Applied to COVID-19, this framework makes it possible to monitor the evolution of disease transmission over time under non-stationary conditions such as those that prevailed during this epidemic. To compare different geographical settings with similar population size, we have chosen to illustrate our approach with data from several regions of France and Ireland.

## Materials and methods

### Modeling a time-varying $R_{eff}$

Our framework is based on three main components: a stochastic epidemiological model embedded in a state-space framework, a diffusion process for each time-varying parameter and a Bayesian inference algorithm based on adaptive particle Markov Chain Monte Carlo (PMCMC) (see S1 Text). The main advantage of the state-space framework is to explicitly consider the observation process. This allows for the unknowns and uncertainty in the partial observation of the disease. The proposed epidemiological model accounts for the transmission characteristic of the COVID-19 and the data features. Fig 1 illustrates model compartments and transition flows between them. The temporal variation in the transmission rate $\beta(t)$ was modeled by making the assumption that it is not driven by specific mechanistic terms but evolves randomly though constrained by the data. We consider that $\beta(t)$ follows a continuous diffusion process:

$$d\log(\beta(t)) = \nu.dB(t) \tag{1}$$

where $\nu$ is the volatility of the Brownian process ($dB$) to be estimated. The use of a Brownian process can be viewed as a non-specific hypothesis for the supposed variation of $\beta(t)$ and the volatility $\nu$ being a regularized factor. Intuitively, the higher the values of $\nu$ the variations in $\beta(t)$ but smaller $\nu$ would induce smoother fluctuations of $\beta(t)$. The logarithmic transformation avoids negative values with no biological meaning.

Based on the SEIR model structure that accounts for the asymptomatic states (see Fig 1 and S2 equations in S1 Text), the $R_{eff}(t)$ can be computed as:

$$R_{eff}(t) = \left( \frac{1 + q_1}{2}.(1 - \tau_A) + q_2.\tau_A \right).\frac{\beta(t)}{\gamma}.\frac{S(t)}{N} \tag{2}$$

where $1/\gamma$ is the average infection duration, $\tau_A$ is the fraction of asymptomatic individuals in the population, $(1-\tau_A)$ the proportion of symptomatic infectious individuals and $q_i$ the reduction in the transmissibility of some infected ($I_2$) and asymptomatics ($A_i$) (see Fig 1 and S2 equations in S1 Text). The effect of vaccination (already implemented in studied regions) is introduced in our model simply by considering its effect on the depletion of susceptibles. The number of "effectively protected vaccinated people", proportional to the number of people vaccinated with one and/or two doses, are removed from the susceptible compartment (see S3 equation in S1 Text).

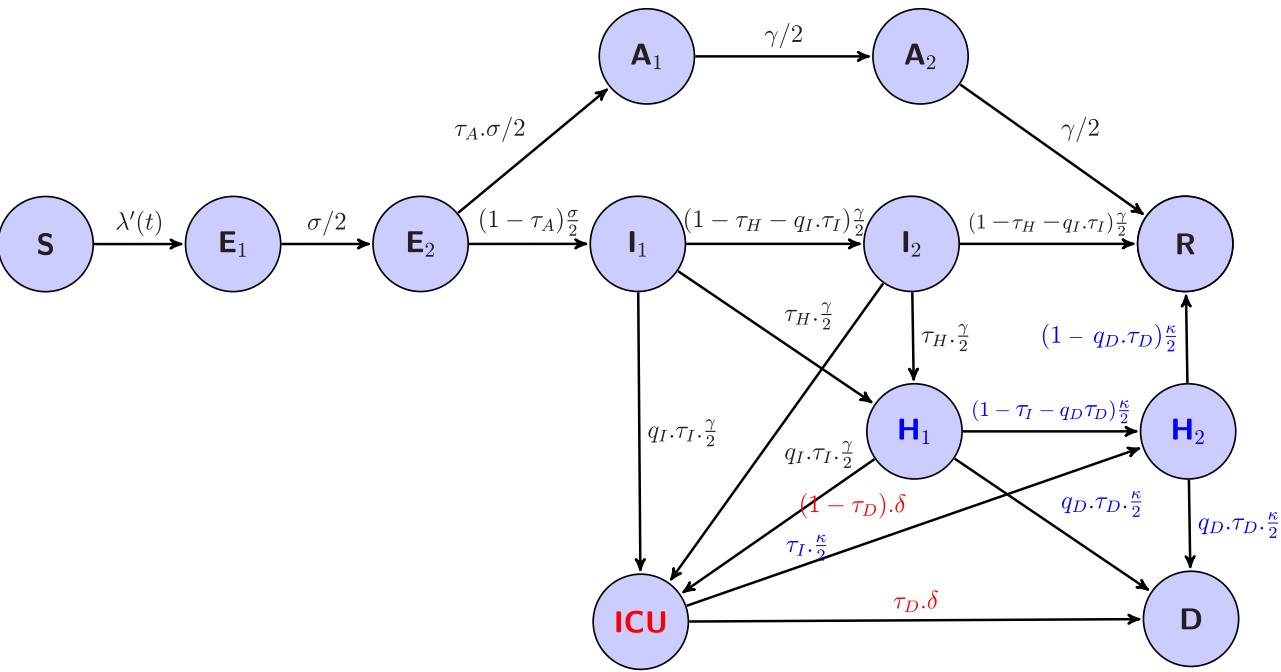

**Fig 1. Flow diagram of a generalized SEIR model accounting for asymptomatic transmission and a simplified hospital system (see S2 equations in S1 Text).** The variables are: the susceptibles $S$, the infected non-infectious $E$, the infectious symptomatic $I$, the infectious asymptomatic $A$, the removed individuals $R$, and the hospital related variables: the hospitalized individuals $H$, the individuals in intensive care unit $ICU$, and the deaths at hospital $D$. The subscripts 1 and 2 stand for the two stages of the Erlang distribution of the sojourn times in $E$, $A$, $I$, $H$. The flow from $H_2$ to $R$ represents hospital discharge. Flows in blue are from hospital ($H_i$) and flow in red from ICU. $\lambda'(t) = \beta(t).(I_1 + q_1.I_2 + q_2.(A_1 + A_2))/N$ then the force of infection is $\lambda(t) = \lambda'(t).S(t)$ and $\beta(t)$ is the time-varying transmission rate (1), $\sigma$ the incubation rate, $\gamma$ the recovery rate, $1/\kappa$ the average hospitalization period, $1/\delta$ the average time spent in ICU, $\tau_A$ the fraction of asymptomatics, $\tau_H$ the fraction of infectious hospitalized, $\tau_I$ the fraction of ICU admission, $\tau_D$ death rate, $q_1$ and $q_2$ the reduction in the transmissibility of $I_2$ and $A_i$, $q_I$ the reduction in the fraction of people admitted in ICU and $q_D$ the reduction in the death rate.

For parameter estimation we used PMCMC [28] suitable for partially observed stochastic non-linear systems (see S1 Text). The implementation is done with the SSM software [26].

Our estimations of $R_{eff}$ were compared, based on data from Ireland, with those obtained with two other methods. The first method, proposed by Cori et al [14] and frequently used in recent studies analyzing COVID-19 data, relies on the number of new infections generated during a given time period and the serial interval distribution and is implemented in the EpiEstim package [29]. In the second method new infections and hence $R_{eff}$ are generated using a simple discrete SIR model fitted with Kalman-filtering tools [30].

## Data used

During the first wave of the COVID-19 epidemic the number of cases reported was very low and associated with large uncertainties [10,12]. This was due, on the one hand, to the testing capacity (RT-PCR laboratory capacity) that was limited and varied greatly during this epidemic [12] and, on the other hand, to the features of this new virus, such as transmission before symptom onset and substantial asymptomatic transmission, which resulted in a low fraction of infected people attending health facilities for testing. This suggests that hospitalization data were likely to be the most accurate COVID-19 related data [9,17]. Thus, we focused on hospital multiple datasets in France and in Ireland. We used incidence data to avoid all the shortcomings associated with the use of cumulative data (see [31]), *i.e.*: daily hospital admissions, daily ICU admissions, daily hospital deaths and daily hospital discharges. We also used cases both in hospital and in ICU. Taking account for the large variability in the daily observations and in

particular for an important weekly component, since the $1^{st}$ of May for French data and the $1^{st}$ of June for Irish data, we have used a weekly average of the observed daily values.

In France, since the beginning of the epidemic, the French regional health agencies (ARS– Agences Régionales de Santé https://www.ars.sante.fr/) have been reporting a number of aggregated statistics. We have also used data made available on the open platform for French public data [32]. In Ireland, this data are published by the Health Protection Surveillance Centre, HPSC and we have used public data [33]. Since the beginning of 2021, these two open platforms have also published the daily vaccination data.

The Irish heath system was down for some days after a cyber attack on the $14^{th}$ of May. Since this date, data availability has been limited. The weekly average values for the $14^{th}$ of May were computed with 6 days and the weekly average values for the $21^{st}$ with just 5 days. Concerning vaccination the values of the "effectively protected vaccinated people" were the same for the $25^{th}$ -$28^{th}$ of May.

Since these hospital related multiple datasets were only available after the implementation of the first panel of mitigation measures in France and in Ireland and that our aim was to estimate $R_{eff}$ before, during and after the implementation of NPI measures, we also used reported incidence data for the period before the implementation of these measures.

## Results

The temporal evolution of both the transmission rate $\beta(t)$ and $R_{eff}(t)$ (Fig 2A) and the fit of the model to the observed data are displayed in Fig 2. Fig 3 shows the time evolution of the $R_{eff}(t)$ (Fig 3A) and the dynamic of the model's unobserved compartments. Another important characteristic of this epidemic is the fact that the peak of daily hospital admission and daily ICU admission are concomitant (Figs 2G, 2H, S3, S6, S9, S12, S15 and S18). This feature has been incorporated in the model by allowing hospitalization or admission to ICU for each stage of the symptomatic infection (Fig 1 and S2 equations in S1 Text).

The capacity of our framework to describe the different types of data, among which some of them are characterized by large noise (within the Paris region for instance) is also illustrated in Fig 2. The main advantage of this framework consists in its ability to reconstruct the time variation of the transmission rate $\beta(t)$ (Fig 2A). Based on the estimation of $\beta(t)$, we can compute the time-variation of $R_{eff}(t)$ (Fig 2A) and thus reconstruct the observed dynamics of the COVID-19 epidemic. Fig 3 provides the results for each component of the epidemic dynamic model for the Paris region. Our main result is described in Fig 4 displaying the temporal evolution of the $R_{eff}$ in the five French regions considered and in Ireland. Our estimates of the initial value of $R_{eff}$ lie in the range [3.0–3.5] in agreement with other estimates [11,34]. The peak of $R_{eff}(t)$ just before the start date of the first mitigation measures is presumably an effect of the model to accommodate diverging trends between reported case data and hospital related data. Then one can note a decrease of about 80% in $R_{eff}$ between the $1^{st}$ of March and the $1^{st}$ of May (during the first lockdown) in all the regions considered (Fig 4 and Table 1). The reduction in the transmission following the second lockdown was smaller, between 40% and 55% in France, but just 20% in Ireland. In France, at the national level, the epidemic showed a slightly increasing plateau before the third wave in April 2021. However, at the regional level and for regions with low seroprevalence, a local third wave was observed. For these waves in Occitanie and Nouvelle Aquitaine a reduction similar to the second wave was observed (Table 1). On the other hand, in Ireland, due to the UK variant, a large wave was observed in January 2021, the reduction of $R_{eff}$ was again significant (>70%) (Table 1). In all these cases, given the temporality of the decline compared to the timing of the NPIs, these sharp decreases seem to be the result of the implementation of the mitigation measures.

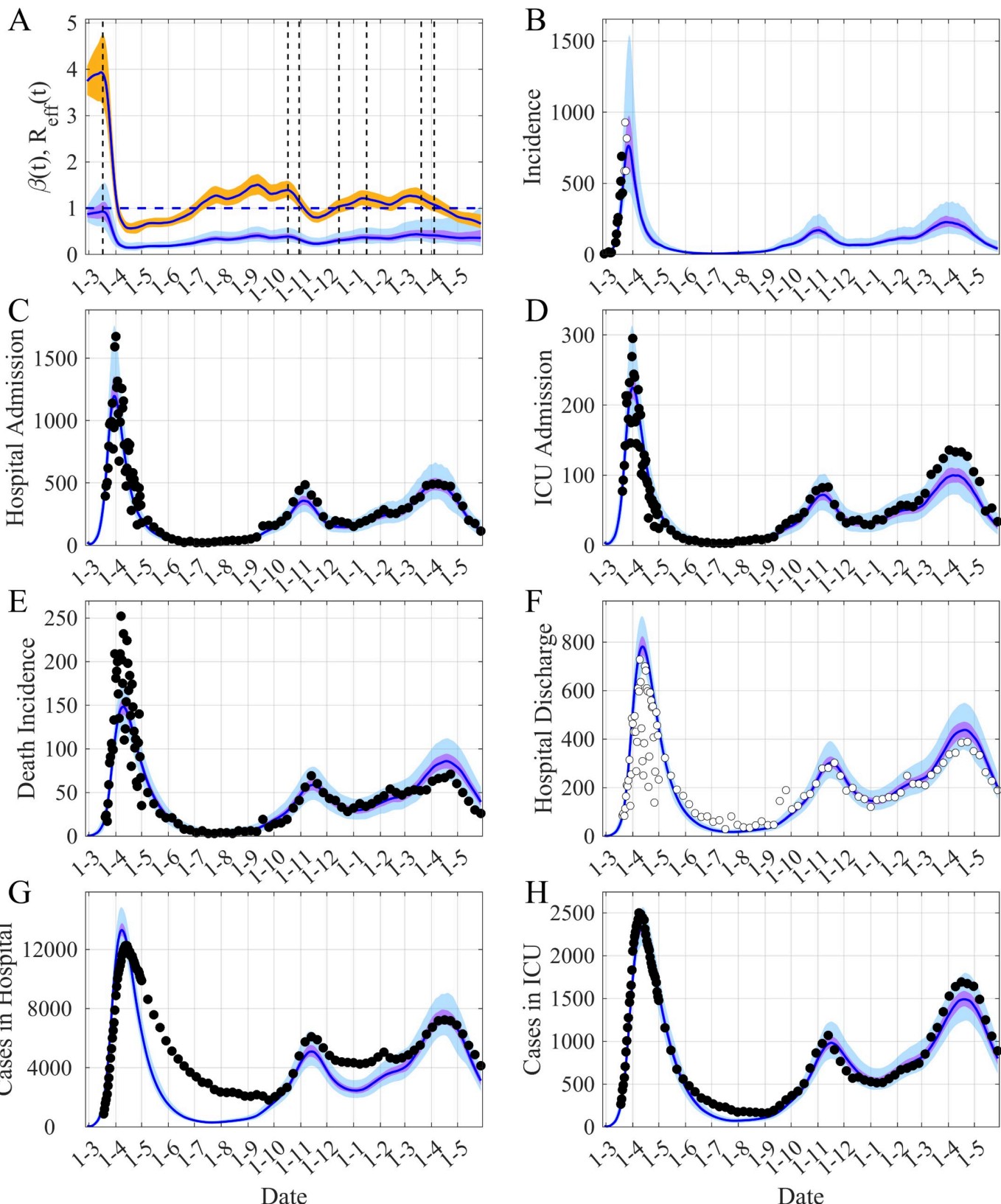

**Fig 2. Reconstruction of the observed dynamics of COVID-19 in Ile-de-France, the Paris region.** (A) Time evolution of both $\beta(t)$ and $R_{eff}(t)$. (B) Simulated and observed incidence. (C-D) New daily admissions to hospital and to ICU. (E) Daily new deaths. (F) Hospital discharges. (G-H) Cases in Hospital and in ICU

per day (average of daily data over the current week is used after 01-05-2020). The black points are observations used in the inference process, the white points are the observations not used. The blue lines are the median of the posterior estimates of the simulated trajectories, the purple areas are the 50% Credible Intervals (CI) and the light blue areas the 95% CI. In (A) the orange area is the 50% CI of $R_{eff}$. The vertical dashed lines show the implementation dates of the main NPI measures and the dot-dashed lines are for cases where only one part of the region has been subjected to these measures. The horizontal dashed-line is the threshold $R_{eff} = 1$. For (B-H), the corresponding reporting rate is applied to the simulated trajectories for comparison with observations.

We can also notice that there is a larger variability at the end of the observation period. As the dynamics of the model are mainly driven by the hospitalization data, these latter determine the transmission rate (and then $R_{eff}$) during the interval $[t\text{-}\delta t, t]$, with $\delta t$ of the order of the average delay between infection and hospitalization, of around 2–3 weeks (Fig 4A and 4B).

Fig 4 also illustrates the robustness and the sensitivity of our method. Firstly, Fig 4A shows that whether hospital discharge data are used or not during the inference process, similar medians and CI are found. Secondly, Fig 4A and 4B highlight the sensitivity to asymptomatic transmission: when asymptomatic transmission increases (i.e. large $q_A$ value) lower values are inferred for $\beta(t)$ hence leading to lower $R_{eff}$ (the opposite happens when $q_A$ decreases).

Fig 3C and 3D show that the number of asymptomatic infectious is of the same order of magnitude as the number of symptomatic infectious but with a larger uncertainty due to lack of information in the data. Indeed, the data used contain very little information on asymptomatics and we observe identical prior and posterior distributions for the rate of asymptomatic, $\tau_A$ (S1, S5 and S8 Figs).

In our model approach, one can also estimate the number of removed individuals. We can clearly show that as the course of the epidemic advances the number of removed individuals increases and this increase is amplified by the introduction of the vaccination (Figs 3H, S4, S7, S10, S13 and S19), inducing a significant rise in seroprevalence (Table 2). Concerning the vaccination, we have only considered an additional depletion term in the susceptibles dynamics (see S1 Text and S2 and S3 equations in S1 Text). The effects of the vaccination on $R_{eff}(t)$ seem not to be yet very significant, the depletion of susceptibles being slightly inversely proportional to an increase of $\beta(t)$ (see S21 Fig). It is also worth noting that the seroprevalence calculated as of 15th of May for France or the 1st of July for Ireland was entirely consistent with the published results of the seroprevalence surveys (Table 2).

The comparison of the performances of our estimations of $R_{eff}$ with those of two others methods is summarized in Fig 5. It is difficult to compare the absolute values of the estimates obtained with the three methods because the true values are unknown and speculative, but we can limit ourselves to comparing the trends given by these methods. The trajectory of $R_{eff}$ over time computed by other methods fall within the range of our 95% CI, which is large because of uncertainties in the transmission rate, asymptomatic transmission and the different delays needed to describe hospital data. We can also note that the 95% CI of $R_{eff}$ obtained with the EpiEstim method is very narrow and its width is smaller than the variability of fluctuations in its median. The main differences between the three estimates relate to the time-lags of the effects of the lockdown, the peaks of $R_{eff}$ and the date of crossing the threshold which is equal to 1. These lags range from one week to more than one month and do not correspond to the early or late lags of the date of crossing 1, when comparing to the estimations provided by our method.

A final important point concerns the observed incidence. We have used the incidence data until the 22 March (Fig 2B, black points; S6, S9, S12, S15 and S18 Figs) in the inference process leading to a median of the posterior distribution of the reporting rate of 2.3% (95% CI: 1.5%-3.3%) for the Paris region (S2 Table). The plotted dynamics of the estimated observed incidence uses this value for the whole trajectory. The values of the reporting rate for the other

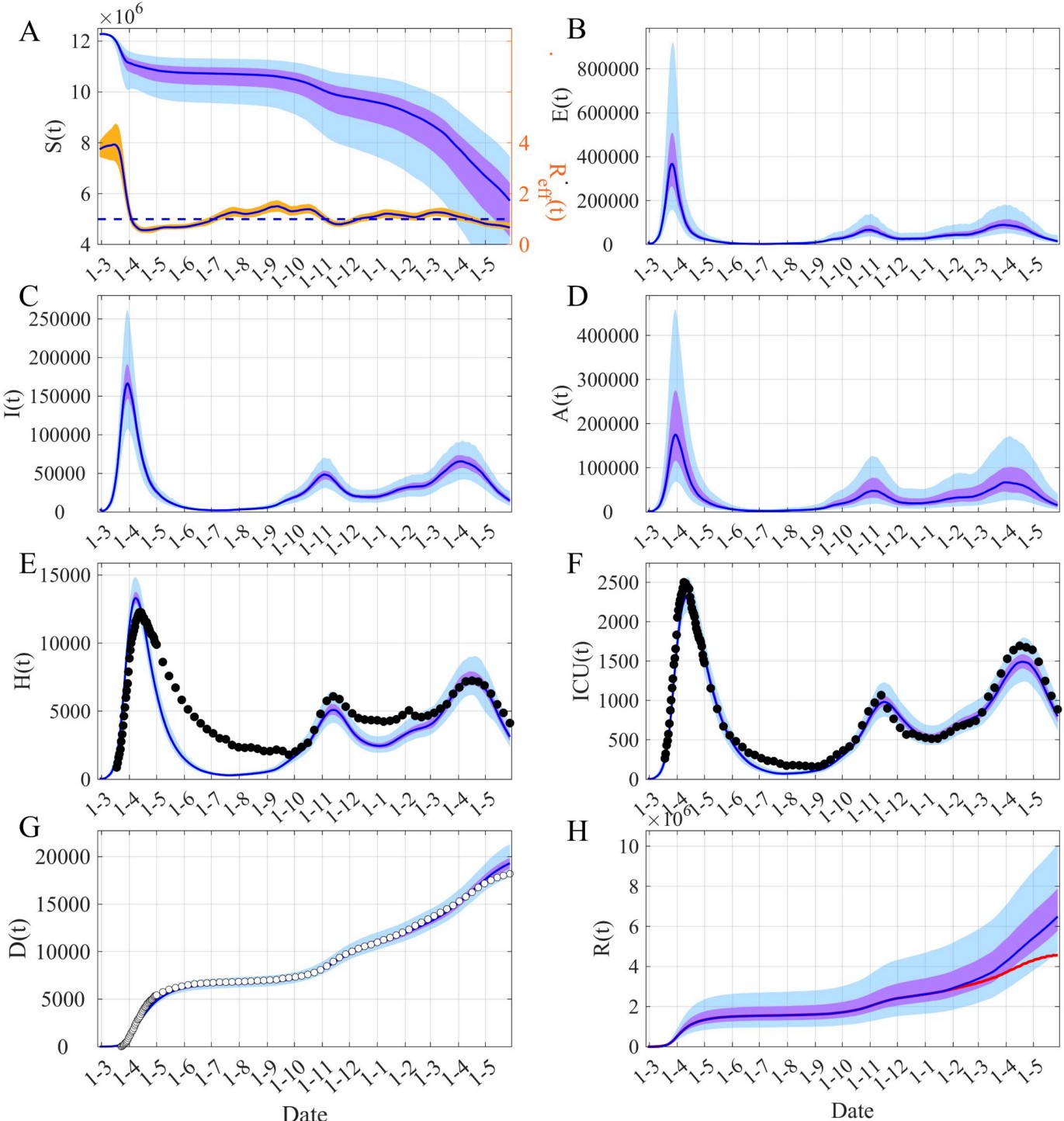

**Fig 3. Model dynamics of COVID-19 in Ile-de-France, the Paris region.** (A) Time evolution of susceptibles $S(t)$ and $R_{eff}(t)$. (B) Infected non infectious, $E(t) = E_1(t)+E_2(t)$. (C) Symptomatic infectious $I(t) = I_1(t)+I_2(t)$. (D) Asymptomatic infectious $A(t) = A_1(t)+A_2(t)$. (E) Hospitalized individuals $H(t) = H_1(t)+H_2(t)+ICU(t)$. (F) Individuals in ICU, $ICU(t)$. (G) Cumulative death $D(t)$. (H) Removed individuals $R(t)$. The blue lines are the median of the posterior estimates of the simulated trajectories, the purple areas are the 50% Credible Intervals (CI) and the light blue areas the 95% CI. In (A) the orange area corresponds to the 50% CI of $R_{eff}$. The black points are observations used in the inference process, the white points are the observations not used. In (H) the red line shows the median of $R(t)$ when the "effectively protected vaccinated people" have been subtracted.

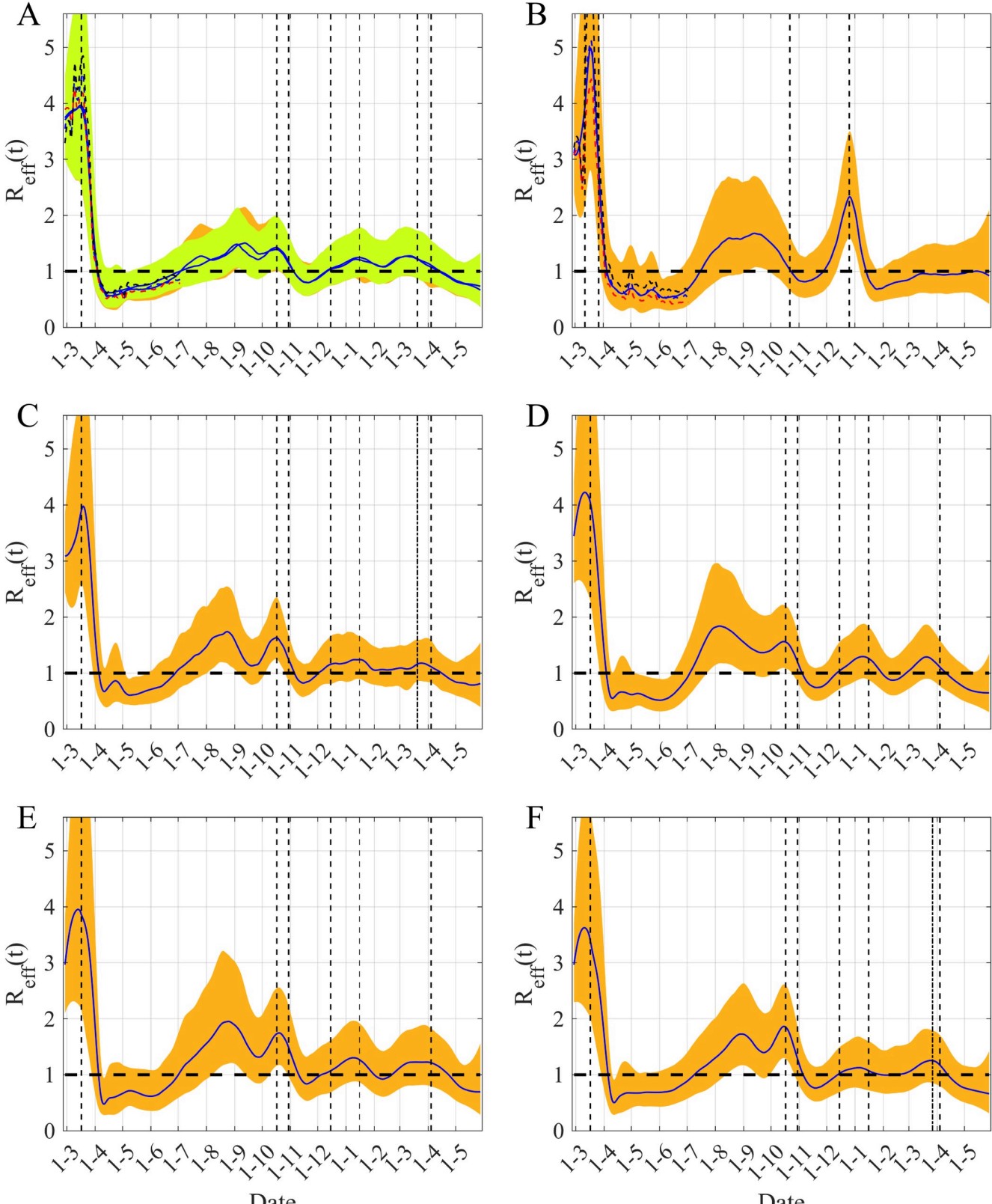

**Fig 4. Time varying $R_{eff}$ (t) in five regions of France and in Ireland.** (A) Ile-de-France, (B) Ireland, (C) Provence Alpes Côte d'Azur, (D) Occitanie, (E) Nouvelle Aquitaine (F) Auvergne Rhône Alpes. The blue lines are the median of the posterior estimates of $R_{eff}$ (t) and orange and yellow areas are the 95%

CI of $R_{eff}$. In (A) the orange area corresponds to the case where hospital discharges are included in the inference process, whereas the yellow area corresponds to the model that does not account for them. In (A) and (B) the dashed curves represent the median of $R_{eff}$ for a preceding time period (blue), or computed with lower transmissibility of the asymptomatics, $q_A = 0.40$ (black) or computed with higher transmissibility, $q_A = 0.70$ (red). The vertical black dashed lines correspond to the start dates of the main mitigation measures, the dot-dashed lines are for cases where only one part of the region has been subjected to these measures. The horizontal dashed-line is the threshold $R_{eff} = 1$.

regions are in S2 Table. The comparison of the observed incidence when available and the simulated incidence (Figs 2, S3, S6, S9, S12, S15 and S18) clearly illustrates that the reporting rate has greatly evolved during the course of the epidemic. It is relatively easy to see that during the first wave the observed peak of incidence comes after the peak of hospitalization, whereas for the second wave the opposite happens, in agreement with what can be expected. Moreover, if we compare the intensity of the observed incidence waves, we can see that the second observed wave is 5 to 10 times greater than the first one, while model-based simulations suggest that they have similar magnitudes. This difference cannot only be explained by the fact that, as the frequency of testing increases, it is also increasingly likely that some of people tested positive are asymptomatic, whereas in the model the people tested are considered symptomatic. In any case, it appears crucial to be able to take into account a reporting delay that seems relatively large for the COVID-19 epidemic using models for now-casting [39,40] or a detailed observation model [18].

## Discussion

For any epidemic, it is always essential to estimate pathogen transmissibility in order to provide information on its potential spread in the population. This is commonly done using the effective reproduction number $R_{eff}(t)$. Its time evolution allows us to follow, almost in real time, the course of the epidemic, and to get information regarding the potential effects of mitigation measures taken.

Here we propose a mechanistic approach based on time-varying parameters embedded in a stochastic model coupled with Bayesian inference. This enables us to reconstruct the temporal evolution of the transmission rate of COVID-19 with the non-specific hypothesis that it follows a basic stochastic process constrained by the available data. Using this approach we can

**Table 1. Estimated $R_{eff}$ reduction for Ile-de-France region, Ireland, Provence Alpes Côte d'Azur region (PACA), Occitanie region (OC), Nouvelle-Aquitaine region (NA), Auvergne Rhône Alpes region (ARA).**

| Regions or Country | $R_{eff}$ reduction between the $R_{eff}$ peak date and 01/05/2020 (peak date) | $R_{eff}$ reduction between 01/03/2020 and 01/05/2020 | $R_{eff}$ reduction between 15/10/2020 and 27/11/2020 | $R_{eff}$ reduction between 26/12/2020 and 01/02/2021 | $R_{eff}$ reduction between 10/01/2021 and 15/02/2021 |
|---|---|---|---|---|---|
| Ile-de-France[*] | 84.5% (15/03/2020) | 83.9% | 39.6% | | |
| Ile-de-France[**] | 83.2% (15/03/2020) | 82.3% | 40.6% | | |
| Ireland[**] | 86.1% (19/03/2020) | 77.5% | 20.8% | 70.7% | |
| PACA[*] | 81.1% (18/03/2020) | 76.8% | 46.0% | | |
| OC[*] | 85.4% (11/03/2020) | 83.2% | 50.8% | | 32.1% |
| NA[*] | 83.4% (12/03/2020) | 79.7% | 46.7% | | 29.8% |
| ARA[*] | 81.4% (10/03/2020) | 78.8% | 56.5% | | |

[*]not using hospital discharge data

[**]using hospital discharge data.

**Table 2. Comparison between seroprevalence from different surveys and our seroprevalence estimations for Ile-de-France region, Ireland and four other French regions: Provence Alpes Côte d'Azur (PACA), Occitanie (OC), Nouvelle-Aquitaine (NA), Auvergne Rhône Alpes (ARA).** For May 2021, we also show our seroprevalence estimations when "effectively protected vaccinated people" are subtracted from the compartment removed.

| Regions or Country | Seroprevalence 05–2020 [35] Median [95%CI] | Seroprevalence 05 to 06–2020 [36] n = 14628 Median [95%CI] | Seroprevalence 05–2020 [37] Median [95%CI] | Seroprevalence 06–2020 [38] Median [95%CI] | Seroprevalence 15-05-2020 *** Median [95%CI] | Seroprevalence 28-05-2021 Median [95%CI] (Median without vaccinated) |
|---|---|---|---|---|---|---|
| **Ile-de-France**[*] | 7.3% [5.5%-9.4%] n = 536 | 10.0% [9.1%-10.9%] | 9.2% [7.1%-11.2%] n = 2350 | | 12.7% [8.0%-21.4%] | 56.1%, [41.2%-80.7%] (39.6%) |
| **Ile-de-France**[**] | | | | | 12.8% [7.7%-19.6%] | 56.5%, [39.4%-78.5%] (39.9%) |
| **Ireland**[**] | | | | 1.7% [1.1%-2.4%] | 2.0% [1.2%-3.6%]*** | 25.8% [22.2%-31.1%] (7.6%) |
| **PACA**[*] | 1.5% [0.7%-2.6%] n = 397 | | 5.2% [2.9%-7.5%] n = 1643 | | 5.2% [3.7%-8.7%] | 57.4% [47.7%-67.0%] (38.0%) |
| **OX**[*] | 1.2% [0.4%-2.4%] n = 206 | | 1.9% [0.9%-2.9%] n = 581 | | 2.5% [1.4%-4.2%] | 41.0% [32.3%-(53.7%) (20.6%) |
| **NA**[*] | 1.4% [0.7%-2.5%] n = 282 | 3.1% [2.4%-3.7%] | 2.0% [0.8%-3.2%] n = 548 | | 1.7% [1.0%-2.6%] | 35.7% [29.3%-41.6%] (13.1%) |
| **ARA**[*] | 2.8% [1.5%-4.5%] n = 234 | | 4.8% [3.3%-6.2%] n = 732 | | 5.3% [3.2%-8.4%] | 53.0% [39.3%-72.1%] (33.8%) |

[*]not using hospital discharge data

[**]using hospital discharge data

[***]01-07-2020 for Ireland.

describe both the temporal evolution of the COVID-19 epidemic and its $R_{eff}(t)$. Thus, we can quantify the effect of mitigation measures on the epidemics waves in France and in Ireland. It is important to note that this methodology overcomes many of the biases overcomes associated with estimates of $R_{eff}(t)$ obtained using existing methods in the case of COVID-19 [6,7]. Indeed, these biases are related to under-reporting of infectious cases, uncertainties in the generation time and in the serial interval, and also to the importance of silent transmission.

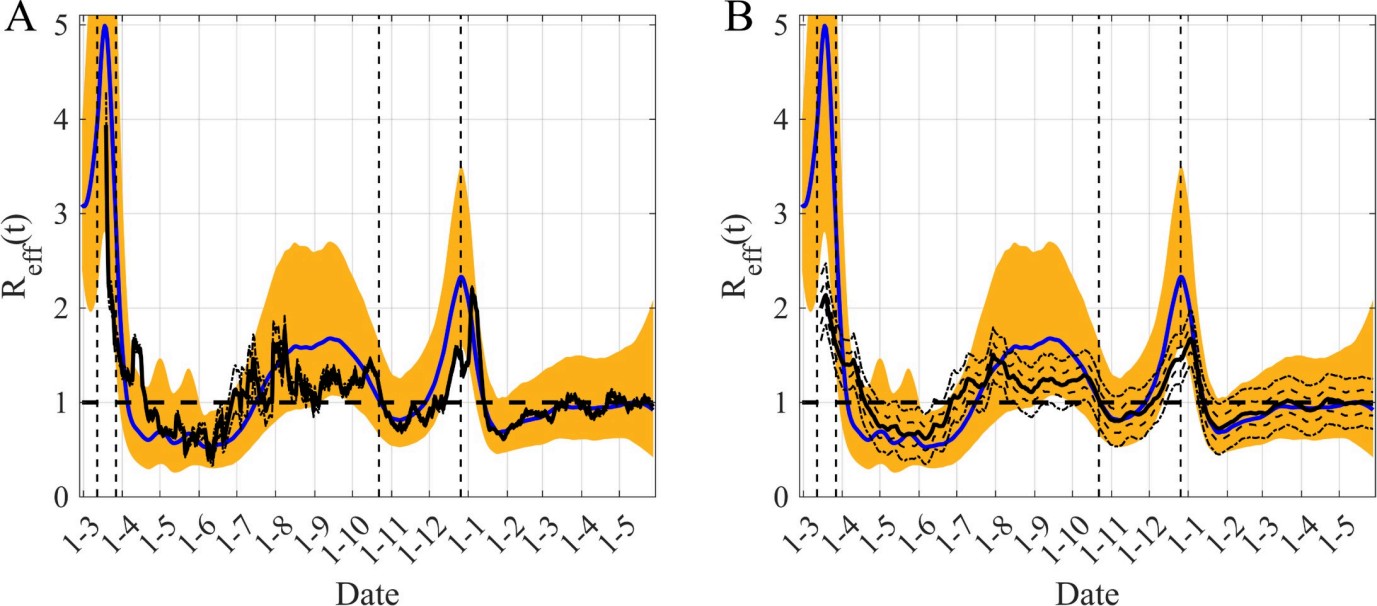

**Fig 5. Comparison between our $R_{eff}(t)$ estimation and those obtained with two other methods based on Irish multiple datasets.** A/ Comparison with the method implemented in EpiEstim R package (http://metrics.covid19-analysis.org/). B/ Comparison with the method proposed by Arroyo-Marioli et al [28] (http://trackingr-env.eba-9muars8y.us-east-2.elasticbeanstalk.com/). Blue lines are the median of the posterior of our estimates of $R_{eff}(t)$ and orange areas are the corresponding 95% CI of our $R_{eff}$ estimates. The black lines represent the median of $R_{eff}(t)$ for the other methods and the black dotted-dashed lines delimit the corresponding 95% CI associated. In B/ the black dashed lines delimited the 65% CI. The vertical black dashed lines correspond to the start dates of the main mitigation measures. The horizontal dashed-line is the threshold $R_{eff} = 1$.

Moreover, these biases are amplified by the fact that reporting delays have fluctuated widely over the course of the epidemic [8].

Within our framework, the estimates of $R_{eff}(t)$ represent the values required to reproduce the well-documented hospital data. Such data are clearly of a higher quality than the reported number of cases [17,18]. Moreover, this estimation accounts for transmission mechanisms and the different delays describe the transmission process (even during the asymptomatic phase) and the different processes related to the hospital (Fig 1). Using our approach, by visualizing the evolution of $R_{eff}(t)$ we can follow the course of the COVID-19 epidemic in five French regions and in Ireland. Therefore, we can quantify the effect of the mitigation measures during and between epidemics waves. For the first lockdown, we estimated a decrease of around 80% in transmission in both countries. For the second wave, our reduction estimations were between 45 to 55% in French regions and around 20% in Ireland (Table 1). While France is undergoing another major epidemic wave, we have estimated a reduction of around 70% of the transmissibility of the third wave in Ireland (Table 1). These reductions in transmission may reflect the nature of the mitigation measures implemented in both countries. For the second wave, these measures were less restrictive than during the first wave, nevertheless the second wave was also less severe. In Ireland, the mitigation measures introduced in the third wave were similar to the first wave.

We also found other interesting results such as a significant high correlation between the trend of mobility and our estimation of the transmission between the epidemic waves (see S22 Fig and [41]), highlighting the importance of following the evolution of mobility when relaxing mitigation measures to anticipate the future evolution of the spread of the SARS-CoV-2.

We have also compared the trend of our estimations of $R_{eff}$ with those of two other methods from the literature [14,30] on data from Ireland (Fig 5). One of the differences between our estimations and the two others is a greater variability of our $R_{eff}$ estimations due to an underlying complex mechanistic model used and to uncertainties in the transmission rate, asymptomatic transmission and in the different delays needed for describing hospital multiple datasets. A second difference consists in asynchronous peaks of $R_{eff}$. For instance, the peak of $R_{eff}$ occurs at the beginning of August for the two other methods, while our estimates suggest a peak in mid-September (Fig 5). These differences may be explained by the differences in model complexity but, above all, by the fact that the two other methods computed their estimations based on the new cases only, which are data subject to certain bias [6–8]. The peak of $R_{eff}$ in early August could be explained by an increase in testing during summer holydays while our estimates peaked later due to the increase in hospitalization generated by higher values of new infections in early September when the economy restarted. We therefore believe that our estimates based on admissions in hospital and ICU and deaths were more consistent with the peak of the second waves of hospitalization that was observed in late October in Ireland (S6 and S7 Figs). This reinforces the relevance of our modeling and inference framework to present a more coherent picture of the evolution of this epidemic.

The main characteristic of our approach is the dependence of the $R_{eff}(t)$ estimations on the relevance of the underlying mechanistic model and the accuracy and completeness of the available data. In our case, inference was based on hospital data that are clearly of a higher quality and accuracy compared to the observed number of infected cases. Moreover, despite its relative simplicity, the model incorporating time varying transmission rate is able to accurately describe the hospital multiple datasets that included daily hospitalized admission for COVID-19, daily ICU admission, daily deaths at hospital and also the number of beds used each day both in hospital and ICU. Furthermore our model can be partially validated by quite a good description of daily hospital discharges that are not explicitly used in the fitting process. This partial validation is strengthened by the fact that our predicted seroprevalence in the French

regions and in Ireland are in complete agreement with the results published from seroprevalence surveys in these settings (see Table 2). To highlight our model results, we can see that the asymptomatic infectious are as numerous as symptomatic ones, but are characterized by a larger uncertainty, due to the lack of information in the data (Figs 3C, 3D, S4, S7, S10, S13, S16 and S19). This is in agreement with recent papers [11,24,42,43], which emphasize that the growth of the COVID-19 epidemic is driven by silent infections. This has also been highlighted in Ireland where it has been estimated that during the second epidemic wave the ratio of silent infections to known reported cases was approximately 1:1 [44]. It is indeed interesting to note that our model estimates for asymptomatic cases in Ireland lead to similar ratios but with a large 95% CI (see S7 Fig).

Our study is not without limitations. The model used here is, like all complex SEIR models developed for COVID-19, non-identifiable. This means that it is likely that several solutions, *i.e.* several sets of parameter values, allow to reproduce observations and we only present one of the most likely ones. This point is overlooked very often but see [45]. One limitation is the use of the classical homogeneous mixing assumption in which all individuals are assumed to interact uniformly and ignores heterogeneity between groups by sex, age, geographical region. However, this kind of data is not readily available. When mixing patterns among age groups are available at the individual level in contact tracing databases, they are only accessible following extensive ethical reviews. Another weakness is related to the absence of an age-structure in the model, which would allow generating age-specific predictions. In all cases, considering an age structure and a contact matrix appears insufficient and heterogeneity of contacts is important (see [46]). Nevertheless, in our opinion, these limitations are more than balanced by the fact that we take into account the non-stationarity of the epidemic data and that our results are mainly driven by hospital related data, which is more accurate and timely than the number of infected cases. As our main objective was to infer global $R_{eff}$, and not to explore age-specific mitigation strategies, the simplification of the age structure appears justified. The corroboration of our findings on the Irish case on proportions asymptomatic individuals with those of others provides further evidence of this [44].

As demonstrated previously, modeling with time varying parameter is an interesting framework for modeling the temporal evolution of an epidemic even if the knowledge about disease transmission is either incomplete or uncertain [27]. Indeed a large part of all the unknowns can be put in the time-varying parameters described by a diffusion process but driven by the observed data. This is exactly what we have been confronted during the COVID-19 pandemic, as the data are uncertain, as are the transmission mechanisms of SARS-CoV-2. We therefore proposed to model the spread of this disease using a stochastic model with a time-varying transmission rate inferred using well-documented hospital multiple datasets. The knowledge of the transmission rate makes it possible to easily calculate the $R_{eff}(t)$, which is a key parameter of the epidemic, in order to monitor the potential effects of public health policies on the course the COVID-19 epidemic. Therefore, we believe that this framework could be particularly useful to analyze the next evolution of the epidemic in relation to the emergence of new variants and to help refine potential mitigation measures, after the third wave, during the period where these measures will be progressively lifted, pending the complete vaccination of the population.

## Supporting information

**S1 Text. Description of model formulation and inference method used, including S1 and S2 Tables.**
(PDF)

**S1 Table. Definition of the different parameters and their priors for Ile-de-France region, Ireland and four other French regions: Provence Alpes Côte d'Azur (PACA), Occitanie (OC), Nouvelle-Aquitaine (NA), Auvergne Rhône Alpes (ARA).**
(PDF)

**S2 Table. Posteriors of the parameters for Ile-de-France region, Ireland, and four French regions: Provence Alpes Côte d'Azur (PACA), Occitanie (OC), Nouvelle-Aquitaine (NA), Auvergne Rhône Alpes (ARA).**
(PDF)

**S1 Fig. Prior and posterior distributions for the model inference presented Fig 2.** $I_1(0)$ is the initial number of infectious individuals, $\nu$ is the volatility of the Brownian process of $\beta(t)$, $1/\sigma$ the average duration of the incubation, $1/\gamma$ the average duration of infectious period, $1/\kappa$ the average hospitalization period, $1/\delta$ the average time spent in ICU, $\tau_A$ the fraction of asymptomatics, $\tau_H$ the fraction of infectious hospitalized, $\tau_I$ the fraction of ICU admission, $\tau_D$ the death rate, $\rho_I$ the reporting rate for the infectious, $\rho_H$ the reporting rate for the hospitalized people.
(TIF)

**S2 Fig. The traces of the MCMC chain for the model inference in Fig 2.** $I_1(0)$ is the initial number of infectious, $\nu$ is the volatility of the Brownian process of $\beta(t)$, $1/\sigma$ the average duration of the incubation, $1/\gamma$ the average duration of infectious period, $1/\kappa$ the average hospitalization period, $1/\delta$ the average time spent in ICU, $\tau_A$ the fraction of asymptomatics, $\tau_H$ the fraction of infectious hospitalized, $\tau_I$ the fraction of ICU admission, $\tau_D$ the death rate, $\rho_I$ the reporting rate for the infectious, $\rho_H$ the reporting rate for the hospitalized people. The acceptance rate is equal to 13.6% and the chain is stationary, ie stationarity is not rejected at the 5% level by the Geweke diagnosis.
(TIF)

**S3 Fig. Reconstruction of the observed dynamics of COVID-19 in Ile-de-France, the Paris region but for the inference process the hospital discharges have been used.** Caption as for Fig 2. The black points are observations used by the inference process, the white points are the observations not used.
(TIF)

**S4 Fig. Dynamics of COVID-19 in Ile-de-France, the Paris region but for the inference process the hospital discharges have been used. Caption as for Fig 3.** The black points are observations used by the inference process, the white points are the observations not used.
(TIF)

**S5 Fig. Prior and posterior distributions for the model inference presented in S3 Fig.** Caption as for S1 Fig.
(TIF)

**S6 Fig. Reconstruction of the observed dynamics of COVID-19 in Ireland.** Caption as for Fig 2 but average daily data of the current week is used after 01-06-2020. The black points are observations used by the inference process, the white points are the observations not used.
(TIF)

**S7 Fig. Dynamics of COVID-19 in Ireland.** Caption as for Fig 3. The black points are observations used by the inference process, the white points are the observations not used.
(TIF)

**S8 Fig. Prior and posterior distributions for the model inferences presented in S6 Fig.** Caption as for S1 Fig.
(TIF)

**S9 Fig. Reconstruction of the observed dynamics of COVID-19 in Provence Alpes Côte d'Azur.** Caption as for Fig 2. The black points are observations used by the inference process, the white points are the observations not used.
(TIF)

**S10 Fig. Dynamics of COVID-19 in Provence Alpes Côte d'Azur.** Caption as for Fig 3. The black points are observations used by the inference process, the white points are the observations not used.
(TIF)

**S11 Fig. Prior and posterior distributions for the model inferences presented in S9 Fig.** Caption as for S1 Fig.
(TIF)

**S12 Fig. Reconstruction of the observed dynamics of COVID-19 in Occitanie.** Caption as for Fig 2. The black points are observations used by the inference process, the white points are the observations not used.
(TIF)

**S13 Fig. Dynamics of COVID-19 in Occitanie. Caption as for Fig 3.** The black points are observations used by the inference process, the white points are the observations not used.
(TIF)

**S14 Fig. Prior and posterior distributions for the model inferences presented in S12 Fig.** Caption as for S1 Fig.
(TIF)

**S15 Fig. Reconstruction of the observed dynamics of COVID-19 in Nouvelle Aquitaine.** Caption as for Fig 2. The black points are observations used by the inference process, the white points are the observations not used.
(TIF)

**S16 Fig. Dynamics of COVID-19 in Nouvelle Aquitaine.** Caption as for Fig 3. The black points are observations used by the inference process, the white points are the observations not used.
(TIF)

**S17 Fig. Prior and posterior distributions for the model inferences presented in S15 Fig.** Caption as for S1 Fig.
(TIF)

**S18 Fig. Reconstruction of the observed dynamics of COVID-19 in Auvergne Rhône Alpes.** Caption as for Fig 2. The black points are observations used by the inference process, the white points are the observations not used.
(TIF)

**S19 Fig. Dynamics of COVID-19 in Auvergne Rhône Alpes.** Caption as for Fig 3. The black points are observations used by the inference process, the white points are the observations not used.
(TIF)

**S20 Fig. Prior and posterior distributions for the model inferences presented in S18 Fig.** Caption as for S1 Fig.
(TIF)

**S21 Fig. Assessment of the effects of "vaccination" on the dynamics of $\beta(t)$ and $R_{eff}(t)$ in Ile-de-France.** For the case with vaccination, the color areas and blues lines are similar to Fig 2. For the case without vaccination, for $\beta(t)$ the red line is its median and the dashed red lines are the 95% CI, for $R_{eff}(t)$ the black line is its median and the dashed black lines are the 50% CI. The same observations between 27-02-2020 and 16-04-2021 have been used for these two cases. The vertical dashed lines show the dates of the implementation of the main NPI measures and the horizontal dashed-line the threshold $R_{eff} = 1$. In (A) the hospital discharges are included in the inference process, whereas in (B) the inference process does not account for them.
(TIF)

**S22 Fig. Parallel trends in effective reproduction number and public transport mobility (https://www.google.com/covid19/mobility/).** (A) Ile-de-France, (B) Ireland, (C) Provence Alpes Côte d'Azur, (D) Occitanie, (E) Nouvelle-Aquitaine, (F) Auvergne Rhône Alpes. Black line: time evolution of the estimated $R_{eff}(t)$ and blue line: public transport mobility. In (A) the back line corresponds to the case where hospital discharges are included in the inference process, whereas the dashed-line corresponds to the model that does not account for them. The vertical black dashed lines correspond to the start dates of the main mitigation measures, the dot-dashed lines are for cases where only one part of the region has been subjected to these measures.
(TIF)

## Acknowledgments

We thank Una Ni Mhaoldhomhnaigh who extracted and managed the data during the lockdown periods and edited some of the previous version of the manuscript.

## Author Contributions

**Conceptualization:** Bernard Cazelles, Elisabeta Vergu, Benjamin Roche.

**Formal analysis:** Bernard Cazelles.

**Funding acquisition:** Bernard Cazelles, Benjamin Roche.

**Investigation:** Bernard Cazelles, Clara Champagne.

**Methodology:** Bernard Cazelles, Clara Champagne, Benjamin Nguyen-Van-Yen.

**Software:** Bernard Cazelles, Clara Champagne, Benjamin Nguyen-Van-Yen.

**Visualization:** Bernard Cazelles.

**Writing – original draft:** Bernard Cazelles, Catherine Comiskey, Elisabeta Vergu.

**Writing – review & editing:** Bernard Cazelles, Clara Champagne, Benjamin Nguyen-Van-Yen, Catherine Comiskey, Elisabeta Vergu, Benjamin Roche.

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
