## [Decision Letter · Decision Letter 0]

12 Apr 2021

Dear Prof. Cazelles,

Thank you very much for submitting your manuscript "A mechanistic and data-driven reconstruction of the time-varying reproduction number: Application to the COVID-19 epidemic" for consideration at PLOS Computational Biology.

As with all papers reviewed by the journal, your manuscript was reviewed by members of the editorial board and by several independent reviewers. In light of the reviews (below this email), we would like to invite the resubmission of a significantly-revised version that takes into account the reviewers' comments.

Please also note the journal's new code sharing policy and ensure that your code is available to the reviewers and editors at this stage and made publicly available prior to publication.

We cannot make any decision about publication until we have seen the revised manuscript and your response to the reviewers' comments. Your revised manuscript is also likely to be sent to reviewers for further evaluation.

Sincerely,

Roger Dimitri Kouyos

Associate Editor

PLOS Computational Biology

Virginia Pitzer

Deputy Editor-in-Chief

PLOS Computational Biology

Reviewer's Responses to Questions

**Comments to the Authors:**

Reviewer #1: In this manuscript, authors propose a bayesian framework to reconstruct the temporal evolution of the transmission rate of COVID-19 pandemic using a stochastic model based on Brownian Motion. Authors did a very comprehensive analysis of their results, and I really appreciate the deep level of investigation they carried out. Although I fully support the publication of this manuscript, I believe that it can benefit from certain changes and clarifications to main text.

— General comments about the paper

- Throughout the text, authors emphasize that their approach has no specific hypothesis on the evolution of the transmission rate. Although they use a diffusion process to model the change in the transmission rate, this does not mean that they do not have any assumptions on it. In fact, they are using a relatively complex SEIR model with many compartments, which intrinsicly imposes assumptions on the dynamics of the disease spread, which are manifested in the parameters defining the flow between these compartments, which are included in Eq. (2).

- Again, regarding the modeling of beta : as far as I see it, there are two things to distinguish here. One is the time-dependent nature of beta, the other is that the derivative of beta being a normal variable. Since the novelty of this manuscript mainly lies in Eq (1), the additional effect of modeling *beta(t)* as Brownian Motion as opposed to a time-independent variable *beta*, or a time-varying *beta(t)* in a different fashion (since there are modeling studies which use beta as a time dependent variable in different ways as well) should be better explained throughout the manuscript. On the same note, I would suggest that authors re-construct the first paragraph of page 4 (starting on page 3) where they mention the advantages of this approach. Considering the asymptomatic transmission or including the mechanisms or pathogen transmission is not specific to this study, nor the novelty in it. The whole idea of using Brownian motion should be better emphasized to distinguish this work than the others which include similar models and time-dependent variable approaches.

- Brownian process is non-staionary, but the increments of it are staionary (as reflected in the volatility parameter in Eq(1)). Can authors comment on how the value of this parameter effect modeling the sudden changes (such as lockdowns imposed overnight) relative to the time-step of the integration process of the ODE system? In the supplementary information, there are upper and lowerbounds defined for this parameter but the reasoning of these bounds are not really discussed. Are those numbers heuristic? (There is a typo on the upperbound, it should be 0.15, but written as 015)

- Why is there no flow between the infectious asymptomatics (A1 and A2) to the infectious symptomatics (I1 and I2) compartments? There is accumulating evidence on pre-symptomatic transmission of COVID-19 as well as asymptomatic - can authors comment on the choice of this way of modeling? On one hand, I realize that this would include even more variables, but the SEIR model is already complex and authors run into identifiabiliy issues already as they mention in the discussion. On the other hand, as authors also mention, there is not enough data to distinguish the asymptomatic versus symptomatic transmission. So I really wonder why authors chose this SEIR structure with this particular complexity.

- There is a confusion about assesing the effect of interventions versus corelating them with the estimated Reff throughout the text. Authors claim that they appropriately asses the effect of mitigation strategies (and then say that they “correlate” certain other NPI with Reff in other parts of the manusctipt). What changes Reff is very multidimensional and depends on many factors, so appropriately assessing any effect by using the changes in Reff in this study is too strong of a claim, since there is no hypothesis on the fact that it will change given the mitigation measures. In that sense, the argument of having a hypothesis-free transmission rate versus using the estimates of it to quantify a certain NPI does sound contradictory (Also mentioned in the discussion, which should be fixed).

- Why data from France and Ireland are used only? I am not asking for more analysis for different countries here, since there is no end to that. But would be useful to know why.

- Authors use the “silent” transmission and “asymptomatic” transmission in many parts of the manuscript. It would be good to define what “silent” transmission means exactly. Do they also include the symptomatic transmissions that are not reported / or asymptomatic transmissions that are reported? What does that really include?

- Regarding the uncertainities in the generation time and the serial interval (Page 3) - do the authors refer to aleatoric (as those variables being random variables with a certain distribution) or epistemic (as the measurement errors and the variability across different studies) uncertainity here?  This should be better explained troughout the text where the word “uncertainity” is used for other variables as well.

- Authors compare their results with 1) results from EpiEstim package, and 2) the study by Arroyo-Marioli et al. Although I appreciate that the results are compared, these two studies use models that have different complexity (one is based on purely the incidence data, and the other is a simpler SIR model compared to the authors’ comples SEIR model). This should be mentioned in the text, since the differences in estimation accuracy cannot be attributed to the Bayesian approach or the Stochastic beta used in this study solely, but is a combination of both the differrences in model complexity, estimation methods, and the incorporated data (In the discussion -> “These differences may be explained by the fact that the two other methods computed their estimations based on the new cases only” - this should be better discussed). On the same note, there are a lot of other studies on estimating Reff - a numerical comparison is not necessary for each, but more of these studies can be included in the main text to give a broader view on the state of the current research to the reader.

- “Taking account of the large variability in the daily observations, since the 1st of May for French data and the 1st of June for Irish data, we have used a weekly average of the observed daily values.” - Why? Isn’t the idea of using Brownian motion for beta is exactly to deal with this variability? Can authors comment on that?

- Discussion (Page 9, Paragraph 1) : “[…] These differences may be explained by the fact that the two other methods computed their estimations based on the new cases only. “ - Given the differences in the number of compartments used in these methods, delays between tthe peaks in different signals will be different - so can’t this difference in asynchronous peaks regarding other studies be attributed to the general structure of the transmission models they use as well?

- Tables S1-S3 : Can authors provide the basis for the upper / lower bounds of the prior distributions they use in their study? Are they adopted from other studies or heuristically defined?

- Fig S2 : It is a better practice to show multiple chains to demonstrate that the chains are mixing properly.

- FIgs S6-S9-S12-S15-S18 : Due to the dynamics of the SEIR model, there is naturally a second peak in the incidence data estimated by the model correlating with the second peaks in other signals (death, hospital, ICU). This second peak is not observed in the data, and it seems like a re-occuring issue for all the regions / countries authors included in their analysis - nevertheless, this is not discussed in the discussion section (only briefly for Fig S6). Can this be attributed to differences in testing only for 5 different regions simultaneously.

- Although not necessary, I would strongly recommend the authors to provide a public repository of their work for reproducibility purposes.

— Wording

- PMCMC is an abbreviation and the full name should be used first before the abbreviation itself.

- […] which measures how many amore people -> how many *more* people (I would recommend to use line numbers next time, makes it easier for both the reviewers and the authors.)

- “multiple hospital datasets” may sound better than “hospital multiple datasets”.

Reviewer #2: in attachment

**Have all data underlying the figures and results presented in the manuscript been provided?**

Reviewer #1: Yes

PLOS authors have the option to publish the peer review history of their article (what does this mean?). If published, this will include your full peer review and any attached files.

Reviewer #1: No

Reviewer #2: No

**Have the authors made all data and (if applicable) computational code underlying the findings in their manuscript fully available?**

Reviewer #2: Yes
---

## [Decision Letter · Decision Letter 1]

3 Jun 2021

Dear Prof. Cazelles,

Thank you very much for submitting your manuscript "A mechanistic and data-driven reconstruction of the time-varying reproduction number: Application to the COVID-19 epidemic" for consideration at PLOS Computational Biology. As with all papers reviewed by the journal, your manuscript was reviewed by members of the editorial board and by several independent reviewers. The reviewers appreciated the attention to an important topic.

The reviewers have a few very minor suggestions, which we would like to give you a chance to address prior to acceptance. Once any changes have been made and submitted, we will accept the manuscript.

Sincerely,

Roger Kouyos

Associate Editor

PLOS Computational Biology

Virginia Pitzer

Deputy Editor-in-Chief

PLOS Computational Biology

[LINK]

The reviewers have a few very minor suggestions, which we would like to give you a chance to address prior to acceptance. Once any changes have been made and submitted, we will accept the manuscript.

Reviewer's Responses to Questions

**Comments to the Authors:**

Reviewer #1: I would like to thank the authors very much for addressing all my questions. The main text is properly adjusted, and in my opinion reads much better now.

I only have one more suggestion - not mandatory but I think it would be better for the sake of scientific rigor. Since we had the discussion on how many markov chains are used for convergence diagnostics, it can be very useful if authors provide the configurations of their simulations in the supplementary material - such as how many chains are used per realization, what is the burn-in period, what is the number of jumps per chain, what is the dilution, and as the authors mention themselves, what are the diagnostics (such as geweke criterion etc.)?

Reviewer #2: Except for a minor comment in the abstract section, as a reader, the current version of the manuscript is according to my expectations, and I recommend it for publication.

In the abstract, please consider making modifications to the following lines:

"We thus can estimate a reduction of more than 80% for the first wave in all the studied regions but a smaller reduction for the second wave when the epidemic was less active. For the third wave in Ireland the reduction was again significant (>70%).”

It would be consistent in writing style as well as more informative for the reader who just reads the abstract only to represent the reduction amount for second wave using a percentage of point estimate just like for the first and third waves.

**Have the authors made all data and (if applicable) computational code underlying the findings in their manuscript fully available?**

Reviewer #1: Yes

Reviewer #2: Yes

PLOS authors have the option to publish the peer review history of their article (what does this mean?). If published, this will include your full peer review and any attached files.

Reviewer #1: No

Reviewer #2: No

Figure Files:

Data Requirements:

Reproducibility:

References:

---

## [Editor Report · Decision Letter 2]

23 Jun 2021

Dear Professor Cazelles,

We are pleased to inform you that your manuscript 'A mechanistic and data-driven reconstruction of the time-varying reproduction number: Application to the COVID-19 epidemic' has been provisionally accepted for publication in PLOS Computational Biology.

Best regards,

Roger Dimitri Kouyos

Associate Editor

PLOS Computational Biology

Virginia Pitzer

Deputy Editor-in-Chief

PLOS Computational Biology

---

## [Editor Report · Acceptance letter]

19 Jul 2021

PCOMPBIOL-D-21-00263R2 

A mechanistic and data-driven reconstruction of the time-varying reproduction number: Application to the COVID-19 epidemic

Dear Dr Cazelles,

I am pleased to inform you that your manuscript has been formally accepted for publication in PLOS Computational Biology. Your manuscript is now with our production department and you will be notified of the publication date in due course.

With kind regards,

Andrea Szabo
